# Low Zinc Alleviates the Progression of Thoracic Aortic Dissection by Inhibiting Inflammation

**DOI:** 10.3390/nu15071640

**Published:** 2023-03-28

**Authors:** Lin Zhu, Peng An, Wenting Zhao, Yi Xia, Jingyi Qi, Junjie Luo, Yongting Luo

**Affiliations:** 1Department of Nutrition and Health, China Agricultural University, Beijing 100193, China; 2Food Laboratory of Zhongyuan, Luohe 462300, China

**Keywords:** thoracic aortic dissection, low zinc, inflammation, vascular smooth muscle cells, phenotypic transition

## Abstract

Vascular inflammation triggers the development of thoracic aortic dissection (TAD). Zinc deficiency could dampen tissue inflammation. However, the role of zinc as a nutritional intervention in the progression of TAD remains elusive. In this study, we employed a classical β-aminopropionitrile monofumarate (BAPN)-induced TAD model in mice treated with low zinc and observed that the TAD progression was greatly ameliorated under low zinc conditions. Our results showed that low zinc could significantly improve aortic dissection and rupture (BAPN + low zinc vs. BAPN, 36% vs. 100%) and reduce mortality (BAPN + low zinc vs. BAPN, 22% vs. 57%). Mechanically, low zinc attenuated the infiltration of macrophages and inhibited the expression of inflammatory cytokines, suppressed the phenotype switch of vascular smooth muscle cells from contractile to synthetic types, and eventually alleviated the development of TAD. In conclusion, this study suggested that low zinc may serve as a potential nutritional intervention approach for TAD prevention.

## 1. Introduction

Aortic dissection (AD) is a life-threatening condition caused by a tear in the intimal layer of the aorta or bleeding within the aortic wall, resulting in the separation (dissection) of the layers of the aortic wall [1]. This disease progresses rapidly and has a high fatality rate [2]. The fatality rate reaches 50.0% within 24 h after the onset of AD and rises to 68.2% after 48 h [3]. Currently, there are no clinically effective drugs to prevent or delay AD progression, which underscores the urgent need for developing effective treatment strategies.

Several clinical and epidemiological studies have indicated a tight association between low serum zinc concentration and the occurrence and mortality of cardiovascular diseases [4,5,6]. One study analyzed the clinical data of 108 patients with aortic aneurysm and found that the zinc content of aortic wall tissue in patients with thoracic aortic aneurysm (TAA) was lower than that in the control group [7]. Consistently, other studies have confirmed that zinc deficiency exists in the aortic wall of patients with aortic aneurysm [8,9]. It was also found that the concentrations of Zn in serum and tissue of AD patients were significantly lower than that of the healthy group [10]. These studies suggested a strong association between reduced zinc levels and the occurrence of AD. However, the role of zinc in the progression of AD remains unclear.

Diet and lifestyle manipulation have become key steps for the primary and secondary prevention of cardiovascular diseases, and nutritional intervention can effectively improve physical condition by maintaining cardiovascular health [11]. As an essential trace element, zinc has a variety of physiological effects on the human body [12]. Notably, animal models of zinc deficiency have confirmed that low levels of zinc induced lymphopenia and compromised cell- and antibody-mediated immune responses in several studies [13,14]. For instance, zinc deficiency can affect the shift of the Th cells’ response to a Th2 predominance, decrease the killing activity of natural killer cells, reduce levels of phagocytosis and intracellular killing in granulocytes, monocytes and macrophages, and diminish the total number of neutrophils [13,14,15,16,17]. Consistently, another study demonstrated that zinc deficiency reduced the production of interferon-gamma, interleukin-2 (IL-2), and tumor necrosis factor-alpha (TNF-α) [18]. The pathogenesis of TAD includes vascular smooth muscle cells (VSMCs) phenotypic transformation and apoptosis, extracellular matrix (ECM) degradation, endothelial dysfunction and immune cell infiltration [19]. In recent years, a large number of clinical and basic studies have demonstrated that inflammatory response is involved in the formation of the TAD [20,21,22]. The AD patients with serious clinical symptoms and developed progression showed higher activities of inflammatory cells in the aortic wall compared with asymptomatic and clinically stable patients [22]. The infiltration of inflammatory cells increases with aggravation during the AD process, and macrophages are the main inflammatory cells in the dissected aorta [23]. Infiltration of macrophages into the aortic wall recruits subsequent immune cells, increases inflammatory response and promotes the development of the AD [24]. Since the accumulation and activation of inflammatory cells within the vascular wall promote aortic weakening and extracellular matrix degeneration [25], and zinc deficiency dampens tissue inflammation; we speculate that low zinc may ameliorate AD by down-regulating inflammation.

Given the potential association between low zinc and AD pathology, this study investigated the effects of low zinc on β-aminopropionitrile monofumarate (BAPN)-induced TAD model to explore whether low zinc could alleviate the process of TAD and be used as a potential nutritional intervention approach.

## 2. Methods and Material

### 2.1. Animals

Three-week-old male C57BL/6N mice were purchased from Beijing Vital River Laboratory Animal Technology (Beijing, China). The animals were kept in individually ventilated cages pathogen-free and allowed to eat food and drink water freely. Weight loss, behavioral changes and feeding or drinking habits were used to assess the health of mice. All animal procedures were approved by the China Agricultural University Laboratory Animal Welfare and Animal Experimental Ethical Inspection.

### 2.2. Experimental Design

Three-week-old male C57BL/6N mice were administrated with BAPN (2 g/kg/day; A3134, Sigma-Aldrich, St. Louis, MO, USA) in the drinking water for 30 days to induce TAD (*n* = 14 per group). The mice in the BAPN group (*n* = 14) were given a normal diet (30 mg Zn/kg; Beijing HFK Bioscience, Beijing, China). The mice in BAPN + low zinc group (*n* = 14) were given a low zinc diet (1 mg Zn/kg; Beijing HFK Bioscience) after BAPN induced one week. The mice in the control group (*n* = 14) were given water without BAPN and a normal diet (30 mg Zn/kg; Beijing HFK Bioscience) for 30 days. The animal experiment ended after BAPN-induced 30 days, followed by the collection of ultrasound images, blood and aortic tissue of mice in each group.

### 2.3. Echocardiography Analysis

Echocardiography analysis was performed using the Vevo 2100 high-resolution imaging system (FUJIFILM Visual Sonics, Toronto, ON, Canada) equipped with an 18- to 38-MHz (MS400, mouse cardiovascular) scan head. The mouse chest was depilated before echocardiography analysis. Then mice were anesthetized with 0.5–1.5% isoflurane in an anesthesia induction chamber. After anesthesia, mice were placed on a homeothermic plate and applied ultrasonic coupling agent to the chest. The left margin of the sternum, the right margin of the sternum, the apex of the heart and the suprasternal fossa of mice were scanned by the scan head, and ultrasound images of the heart were collected after BAPN-induced 30 days. The maximal diameters of the aorta were measured at the level of the aortic outflow tract of the innominate artery.

### 2.4. Enzyme-Linked Immunosorbent Assay (ELISA)

After the animal experiment ended, the whole blood was collected, left at room temperature for 3 h, centrifuged at 3000 rpm for 10 min, and the upper serum was taken for the ELISA experiment. The levels of TNF-α were estimated with Mouse TNF-α ELISA Kit (EK282/4-01, Multi Sciences, Shanghai, China) according to the manufacturer’s instructions. Briefly, the standards and samples were added into holes of the enzyme label plate, respectively. The TNF-α in the sample was bound to the solid antibody after incubation. Then unbound biotinylated antibody was washed and removed, and horseradish peroxidase-labeled streptavidin (Streptavidin-HRP) solution, TMB substrate solution, and stop solution were gradually added. Absorbance at 450 nm was estimated within 30 min.

### 2.5. Immunofluorescence Staining

Mouse aortic tissues were stored at 4% paraformaldehyde after the animal experiment ended and divided into 5-µm-thick serial sections. Tissue paraffin sections were dewaxed first, and microwave antigen repair was performed. Seal with 5% goat serum (dilute with 1× PBS) after several washes with PBS. Then paraffin sections were incubated overnight at 4 °C with the specific primary antibody Mac-3 (1:100 dilutions; 108501, BioLegend, San Diego, CA, USA). After several washes with PBS, the paraffin sections were incubated with fluorescent secondary antibody Alexa Fluor 488 goat Anti-Mouse IgG (1:1000 dilution; ab150113, Abcam, Cambridge, UK) for 1–2 h at 4 °C. Antifade mounting medium with DAPI (S2110, Solarbio, Beijing, China) was used for the paraffin section’s seal. Immunofluorescence was visualized using a confocal microscope (Zeiss, Jena, Germany).

### 2.6. Hematoxylin and Eosin (HE), Elastic Van Gieson (EVG), Masson’s Trichrome and Alcian Blue Staining

Mouse aortic tissues were divided into 5-µm-thick serial sections, and the paraffin sections were stained after deparaffinized. Aortic sections were stained with hematoxylin and eosin (G1120, Solarbio), elastic Van Gieson (G1593, Solarbio), Masson’s trichrome (G1340, Solarbio), alcian blue (G1560, Solarbio) according to the manufacturer’s instructions. Images were acquired by a fluorescence microscope bright field camera (Leica, Wetzlar, Germany).

### 2.7. Real-Time Quantitative PCR (Q-PCR)

Frozen aortic tissues were homogenized using an automatic bead homogenizer. Total RNA was extracted from aortic tissue using TRIzol (15596026, Thermo, Waltham, MA, USA) according to the manufacturer’s instructions. RNA samples (1 μg) were reverse-transcribed into cDNA with a HiScript III RT SuperMix for qPCR kit (R323-01, Vazyme, Nanjing, China), and the cDNA was amplified by real-time quantitative PCR using SYBR qPCR Master Mix (Q711-02, Vazyme). The amount of target mRNA in samples was estimated by the 2^−ΔΔCT^ relative quantification method. All samples were amplified using at least 3 technical replicates per sample. The primers used for Q-PCR are shown in Appendix A.

### 2.8. Western Blot

Frozen aortic tissues were homogenized using an automatic bead homogenizer. Total proteins were obtained from aortic tissues using a RIPA buffer containing a stop protease and phosphatase inhibitor (9806, Cell Signaling Technology, Danvers, MA, USA). The whole protein samples of each group were heated at 95 °C for 5 min, then equal amounts of protein were loaded and separated into 15%, 10% or 8% sodium dodecyl sulfate-polyacrylamide gels. Subsequently, all proteins were transferred onto the polyvinylidene fluoride membrane. The membrane was incubated overnight at 4 °C with the specific primary antibody; then, they were incubated with the corresponding secondary antibodies. Protein bands were visualized by using a chemiluminescent reagent (Thermo, Pierce, MA, USA) and ChemiScope 3600 MINI (Clinx Scientific Instrument, Shanghai, China). All antibodies used for the western blot are shown in Appendix A.

### 2.9. Statistical Analysis

Data presented were expressed as the mean of three or more biological replicates/biologically independent experiments. Results are analyzed in GraphPad Prism 8.0 (GraphPad Software, San Diego, CA, USA). Differences were analyzed by one-way, two-way analysis of variance (ANOVA) and Tukey’s post hoc test (experiments with ≥3 groups) as appropriate. For survival curves, differences were analyzed with the log-rank (Mantel-Cox) test. Statistical significance was assigned at * *p* < 0.05, ** *p* < 0.01 and *** *p* < 0.001.

## 3. Results

### 3.1. Low Zinc Significantly Mitigates BAPN-Induced TAD Development in Mice

Previous studies have shown that TAD models can be established by administering BAPN to mice [26,27,28]. The mechanism of the BAPN-induced TAD model was that BAPN inhibits the activity of lysyl oxidase (LOX), which catalyzes the cross-linking of lysine residues in elastin and collagen, thereby increasing the degradation of ECM proteins as crucial components of aortic integrity, eventually resulting in the occurrence of AD [29]. To investigate the role of zinc in AD, we performed in vivo experiments using a BAPN-induced TAD mouse model [26,27,28]. The normal zinc (BAPN group) and the low zinc (BAPN + low zinc group) mice were treated with BAPN for 30 days, while the control mice (control group) were treated without BAPN (*n* = 14 per group) (Figure 1A). The BAPN + low zinc group was given a special diet after BAPN was induced for one week. The survival curve showed that low zinc treatment could significantly increase the survival rate compared with the BAPN group (*p* = 0.0249, Figure 1B). During the 30 days of BAPN administration, 57% (*n* = 8) of the BAPN group mice and 22% (*n* = 3) of the BAPN + low zinc group mice died from AD and rupture (Figure 1B,C). In addition, 43% (*n* = 6) of the BAPN group mice and 14% (*n* = 2) of the BAPN + low zinc group mice experienced AD without rupture (Figure 1C). The BAPN group also presented a great difference in aortic diameter and TAD occurrence compared with the BAPN + low zinc group (Figure 1D).

Furthermore, vascular ultrasound images and maximal aortic diameter measurement after 30 days of modeling demonstrated that the BAPN + low zinc group mitigated BAPN-induced aortic dilation and reduced the average maximal aortic diameter in comparison with the BAPN group (Figure 2A,B).

The degradation of ECM during AD progression results in VSMC loss and contributes to rupture and dilation of the aortic wall [30]. Elastic fibers and collagen fibers are two major macromolecules within the arterial ECM. We found that total elastin content and elastic fiber cross-links were reduced during the occurrence of AD, and the collagen expression was increased, and disorderly deposition may correspond to a slow reparative process triggered by elastic fiber fragmentation and depletion [31]. In addition, proteoglycan content is minimal in normal vascular tissue while significantly increasing in diseased vascular tissue [32]. Therefore, we performed a pathological staining analysis. HE staining showed that BAPN-induced dissecting aneurysm formation was alleviated in the BAPN + low zinc group compared with the BAPN group (Figure 3A). EVG staining demonstrated that BAPN-induced elastic fiber fragmentation and disarray were also mitigated in the low zinc treatment group (Figure 3A). BAPN-induced mice caused the characteristic features of ECM degradation, including excessive collagen deposition and proteoglycan accumulation. Masson’s trichrome and Alcian blue staining showed that these features were alleviated by low zinc treatment (Figure 3A). Consistently, the quantitative statistics of elastin breaks and collagen content also indicated that low zinc could mitigate TAD development (Figure 3B,C).

### 3.2. Low Zinc Inhibits the Transition of Contractile VSMCs to Synthetic VSMCs

The characteristic plasticity of VSMCs is that they can adapt to environmental stimuli and mechanical stresses, thereby switching between contractile and synthetic types [33]. During AD progression, the balance between contractile and synthetic VSMCs is shifted towards synthetic VSMCs, and the proteolytic enzyme production is increased [30]. In fact, the phenotypic switch of VSMCs is the key to AD and rupture [34,35]. Thus, the real-time Q-PCR was performed to discern differently expressed genes in the aortic tissues of each group and revealed that VSMC phenotypic switch was inhibited in the BAPN + low zinc group. 

The VSMC contractile genes, including *Myh11* (myosin heavy polypeptide 11), *Acta2* (actin alpha 2), *Myl9* (myosin light polypeptide 9), *Ccn1* (calponin 1) and *Ramp1* (receptor activity modifying protein 1) were up-regulated in low zinc aortas compared with the BAPN group (Figure 4A). Meanwhile, the VSMC synthetic genes, including *Col1A1* (collagen type I alpha 1), *Cxcl2* (C-X-C motif chemokine ligand 2) and *Fn1* (fibronectin 1), were down-regulated in low zinc aortas compared with the BAPN group (Figure 4B). Furthermore, the expression levels of selected typical VSMC contractile proteins (ACTA2, CCN1 and MYH11) were also rescued with low zinc treatment (Figure 4C). Taken together, these results demonstrated the critical role of low zinc in VSMC phenotypic switch in TAD pathogenesis.

### 3.3. Low Zinc Alleviates TAD Development by Reducing Inflammation

Inflammation was involved in the occurrence of TAD by regulating the homeostasis of the aortic wall, and the increase of inflammatory response resulted in VSMCs apoptosis and ECM destruction in the aortic wall [24]. The characteristic of inflammatory response is the presence of T lymphocytes, macrophages, mast cells, and neutrophils. At the onset of AD, there is a large number of infiltrated macrophages into the aortic wall lesions [23]. In our results, the immunofluorescence staining demonstrated that the green fluorescence signal intensity (Mac-3, biomarker of macrophages) decreased significantly undergo low zinc, suggesting low zinc countered BAPN-induced inflammatory cell infiltration of macrophages (Figure 5A,B).

Multiple inflammatory factors, such as tumor necrosis factor-α (TNF-α) and interleukin-6 (IL-6), are released during the occurrence of AD [36]. Enzyme-linked immunosorbent assay (ELISA) showed that TNF-α expression was significantly decreased in aortic tissues of the BAPN + low zinc group compared with the BAPN group (Figure 5C). Q-PCR indicated that transcriptional levels of the tumor necrosis factors, including *TNF-α*, *TNFAIP3*, *TNFSF10*, *TNFSF9* and *TNFSF1A,* were down-regulated in aortas of the BAPN + low zinc group compared with the BAPN group (Figure 5D).

Specific cytokines and chemokines can promote the recruitment of inflammatory cells in the aortic media. As previously reported, the expression of *TNF-α*, *CRP*, *IL-2*, *IL-1β*, *IL-6*, *IL-8* and *MCP-1* (monocyte chemotactic protein-1) is up-regulated in AD, while the expression of *IL-10* is down-regulated [23,37,38,39]. Our data demonstrated that the interleukin and chemokines, including *IL-1β*, *IL-6* and *MCP-1,* were down-regulated in the aortas of the BAPN + low zinc group compared with the BAPN group, while the *IL-10* was on the reverse trend (Figure 5E). Furthermore, the protein levels of selected typical proinflammatory factors (IL-1β, TNF-α and IL-6) were also down-regulated with low zinc treatment (Figure 5F).

In addition, ECM enzymes can promote the degradation of ECM. According to previous studies, the expression of *MMP-2* and *MMP-9* (MMP, matrix metalloproteinase) is up-regulated in AD, and the ADAMTS degrades ECM components in the cardiovascular system [40,41]. Our results showed that the ECM enzymes, including *MMP-2*, *MMP-9* and *ADAMTS,* were down-regulated in the aortas of the BAPN + low zinc group compared with the BAPN group (Figure 5G). Taken together, these results indicated that low zinc significantly reduced transcriptional levels of representative biomarkers during the process of TAD, such as tumor necrosis factors, interleukin and chemokines, and ECM enzymes, eventually leading to alleviating the development of TAD.

In conclusion, low zinc treatment down-regulated the degree of the inflammatory response by decreasing the infiltration of macrophage, reduced the expression of cytokines and chemokines from these inflammatory cells, and inhibited the phenotype switch from contractile VSMCs to synthetic VSMCs, eventually alleviating TAD development (Figure 6). 

## 4. Discussion

Our findings suggest that low zinc treatment can alleviate TAD progression, which is supported by a BAPN-induced TAD model treated with low zinc. To our knowledge, this study is the first to investigate the role of low zinc in TAD progression through nutritional intervention. The evidence, such as the survival curve, the vascular ultrasound and pathological images, and the markers of the VSMC phenotypic switch, indicates a successful construction of the TAD model. Strikingly, it was observed that low zinc significantly improved the formation and rupture of AD and reduced mortality. Further analysis reveals that low zinc down-regulates aortic inflammation by attenuating the infiltration of macrophages, suppressing the switch of VSMCs from contractile to synthetic phenotypes, and eventually inhibiting TAD development. This study not only delineates the critical role of low zinc in TAD progression but also identifies a potential nutritional intervention strategy for TAD prevention through low zinc.

Zinc is an essential trace element of the human body and plays an irreplaceable role in the physiological and biochemical processes [42]. This micronutrient has to be supplied with food on a daily basis to maintain zinc homeostasis and proper function [43]. A zinc-deficient mouse model can be established by feeding mice with a low zinc diet [44,45]. Functional studies have indicated that abnormal trace elements may be involved in the occurrence and development of aortic diseases [10,46,47,48]. For instance, zinc levels were dysregulated in the VSMCs of hypertensive patients [12]. The results of the meta-analysis support the conclusion of zinc reduction in aortic aneurysm patients, although this conclusion still needs to be supported by high-quality evidence [48]. In addition, this study also found that plasma zinc concentrations in patients with an aneurysm were reduced [48]. Therefore, the potential role of low zinc in TAD progression is noteworthy. 

There are two main steps during the formation of AD: (1) aortic intima tear or ulceration, aortic intima rupture, and blood entering the tunica media; (2) rupture of the nourishing vessels in the aortic media, and the blood in the vessels spills into the media. After the formation of dissection, aortic inflammation triggers the dilation and subsequent rupture of the aorta [49]. The inflammatory cells infiltrate into the adventitia of the aortic wall, where they produce enzymes, cytokines and chemokines that can degrade the ECM of the aorta and directly interact with VSMCs [20,50,51,52]. Eventually, the inflammatory response resulted in vascular remodeling and progressive degeneration of the aortic wall, which may finally lead to AD and rupture [53]. We demonstrated that low zinc treatment dramatically decreased the infiltration of inflammatory cells, reduced the expression of inflammatory markers, and mitigated TAD progression. These results suggested low zinc prevented the development of AD by mitigating inflammatory response in the aortic wall.

VSMCs play an indispensable role in vascular homeostasis and contractility [30]. During AD progression, VSMCs convert from a contractile to a synthetic phenotype [34,54], leading to the degradation of ECM, thereby promoting SMC detachment from ECMs and accelerating SMC migration and apoptosis [30]. Here, we demonstrated that low zinc significantly reduced the expression of VSMCs synthetic markers while upregulated the expression of contractile markers, which alleviated the vascular malfunction caused by BAPN induction. Taken together, these data suggest that low zinc-reduced tissue inflammation might alleviate TAD progression partially by inhibiting the synthetic phenotype of VSMCs. However, the molecular mechanism of how low zinc inhibits the phenotypic transition of VSMCs remains unclear.

Apart from the indispensable role of VSMCs in vascular homeostasis, the ECM is critical for the maintenance of the structure and function of the arterial wall by providing elasticity and distensibility. In addition, the ECM also provides critical signals, both directly by interacting with adhesion molecules and indirectly as a reservoir for signaling factors [55]. The ECM degradation and the expression of ECM enzymes such as MMP-2, MMP-9 and ADAMTs are elevated during the development of the AD [39,40,51]. In our study, low zinc down-regulated the expression of some representative ECM enzymes (MMP-2, MMP-9 and ADAMTs) and alleviated the characteristic features of ECM degradation (less fragmentation of elastin fibers, collagen deposition and proteoglycan accumulation). In line with these studies, our results suggested that low zinc intervention inhibited the degradation of ECM in the BAPN-induced TAD model. However, it still needs more studies to explore the potential mechanism.

Although our study reveals the role of low zinc in the process of TAD through in vivo experiments, it is still unclear whether low zinc plays the same role in cell experiments in vitro. In fact, our results suggest that it is difficult to conduct in vitro experiments because the regulation of low zinc on AD is complex, involving immune cells, inflammation and related factors, and SMCs. In the body, zinc homeostasis depends on three important protein families: zinc transporters (ZnTs), zinc importers (ZiPs) and metallothionein (MT) [56]. Whether the constructed conditional zinc transporter/zinc storage protein gene knockout mice have a more positive inhibitory effect on TAD development than low zinc treatment alone needs further experimental exploration. It has been found that zinc levels are reduced in the arteries and serum/plasma of patients with AD based on available clinical data [7,8,9,10]. However, it is unclear whether the incidence of AD in zinc deficiency patients is lower than that in the general population, which warrants further investigation in the future.

Collectively, this study demonstrated the protective effects of low zinc on the development of TAD. Low zinc treatment robustly reduced the infiltration of macrophages and down-regulated aortic inflammation, thereby inhibiting the phenotype switch of VSMCs and the degradation of ECM, leading to diminished TAD progression. In conclusion, our findings suggest that low zinc may be an effective and promising strategy for the prevention and treatment of TAD.

## Figures and Tables

**Figure 1 nutrients-15-01640-f001:**
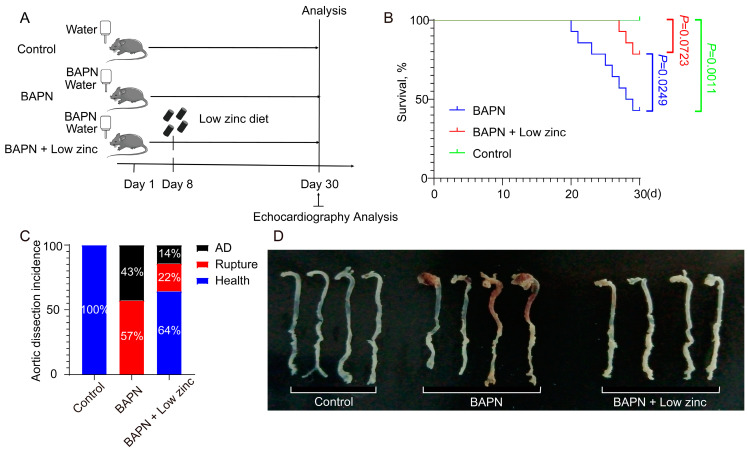
Low zinc represses TAD progression. (**A**) Experimental design diagram. (**B**) The survival rate was estimated by the Kaplan-Meier method and compared by log-rank test (*n* = 14 per group). (**C**) TAD incidence. (**D**) Representative macrographs of the aorta.

**Figure 2 nutrients-15-01640-f002:**
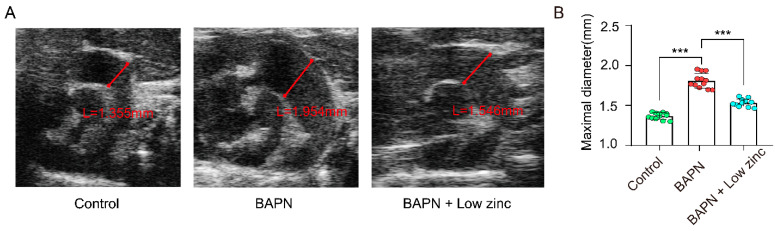
Echocardiography analysis at the end of the animal experiment (BAPN-induced 30 days)**.** (**A**) Representative ultrasound images of thoracic aortas. (**B**) Measurements of maximum aortic diameter, green blots represent control group, red blots represent BAPN group and blue blots represent BAPN + Low zinc group, *n* = 12 per group, by one-way ANOVA (*** *p* < 0.001).

**Figure 3 nutrients-15-01640-f003:**
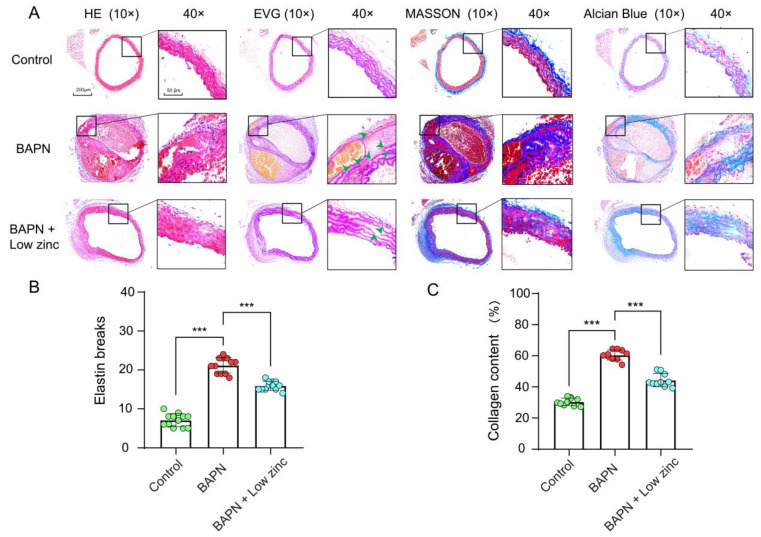
Pathological staining analysis after animal experiment end (BAPN-induced 30 days). (**A**) Representative images showing hematoxylin and eosin (HE) (first column), elastic van Gieson (EVG) (second column), Masson’s trichrome (MASSON) (third column) and Alcian blue (final column) staining in paraffin sections from the indicated mice (top for the control group, middle for the BAPN group, bottom for the BAPN + low zinc group). Green arrowheads indicate elastin breaks. Scale bar, 200 μm for 10 magnifications, 50 μm for 40 magnifications. (**B**) Quantification of elastin breaks in paraffin sections from the mouse cohorts shown in (**A**), *n* = 12per group, by one-way ANOVA (*** *p* < 0.001). (**C**) Quantification of collagen content in paraffin sections from the mouse cohorts shown in (**A**), *n* = 10 per group, by one-way ANOVA (*** *p* < 0.001). Green blots represent control group, red blots represent BAPN group and blue blots represent BAPN + Low zinc group.

**Figure 4 nutrients-15-01640-f004:**
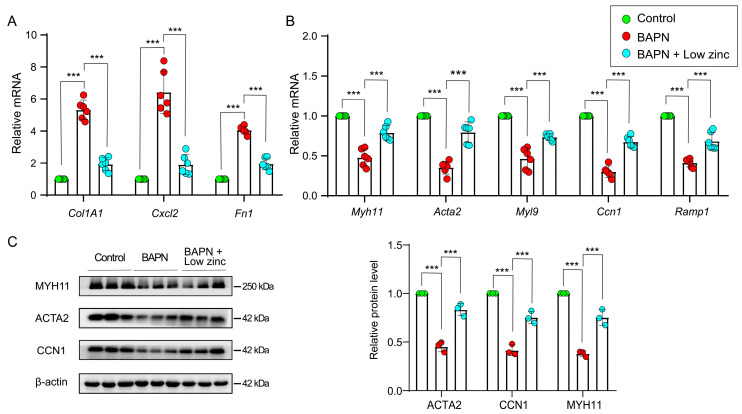
Low zinc suppressed the phenotype switch of vascular smooth muscle cells. (**A**) Relative mRNA levels of selected VSMC synthetic genes (*Col1A1*, *Cxcl2* and *Fn1*) in mice aorta (*n* = 6 per group); by two-way ANOVA (*** *p* < 0.001). (**B**) Relative mRNA levels of selected VSMC contractile genes (*Myh11*, *Acta2*, *Myl9*, *Ccn1* and *Ramp1*) in mice aorta (*n* = 6 per group); by two-way ANOVA (*** *p* < 0.001). (**C**) The protein levels (left column is western blot result, right column is quantitative analysis) of selected VSMC contractile markers (MYH11, ACTA2 and CCN1) in mice aorta (*n* = 3 per group); by two-way ANOVA (*** *p* < 0.001). All samples were collected after the animal experiment ended (BAPN-induced 30 days).

**Figure 5 nutrients-15-01640-f005:**
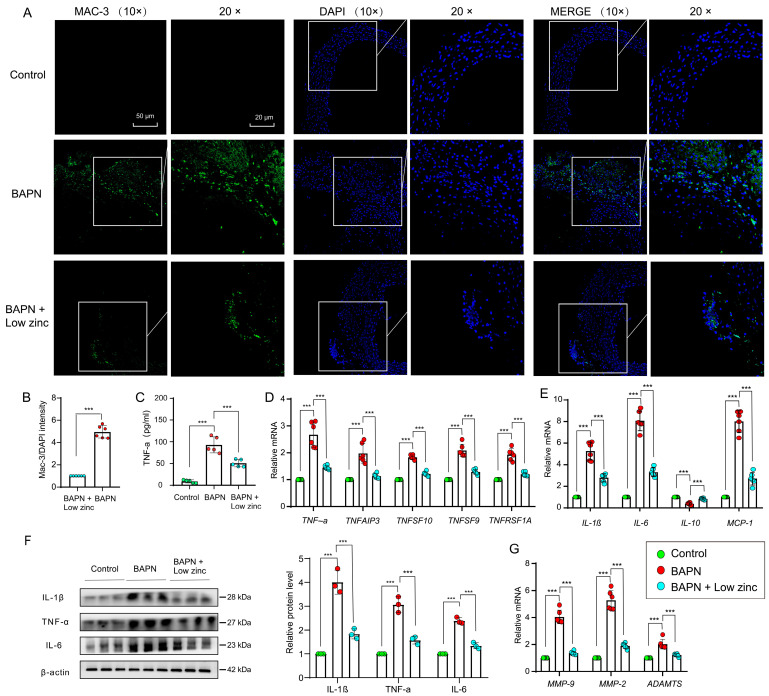
Low zinc inhibited TAD by down-regulating inflammation. (**A**) Representative confocal images of mice aorta stained with Mac-3 (green) and DAPI (blue) (scale bars, 50 μm for 10 magnifications; 20 μm for 20 magnifications). (**B**) Quantification of Mac-3 levels (*n* = 6 per group); by one-way ANOVA (*** *p* < 0.001). (**C**) The expression level of TNF-α in mice serum (*n* = 5 per group), by one-way ANOVA (*** *p* < 0.001). (**D**) Relative mRNA levels of *TNF-α*, *TNFAIP3*, *TNFSF10*, *TNFSF9* and *TNFSF1A* of tumor necrosis factors in mice aorta (*n* = 6 per group); by two-way ANOVA (*** *p* < 0.001). (**E**) Relative mRNA levels of *IL-1β*, *IL-6*, *IL-10* and *MCP-1* of interleukin and chemokines in mice aorta (*n* = 6 per group); by two-way ANOVA (*** *p* < 0.001). (**F**) The protein levels (left column is western blot result, right column is quantitative analysis) of selected typical proinflammatory factors (IL-1β, TNF-α and IL-6) in mice aorta (*n* = 3 per group); by two-way ANOVA (*** *p* < 0.001). (**G**) Relative mRNA levels of *MMP-9*, *MMP-2* and *ADAMTS* of extracellular matrix enzymes in mice aorta (*n* = 6 per group); by two-way ANOVA (*** *p* < 0.001). All samples were collected after the animal experiment ended (BAPN-induced 30 days).

**Figure 6 nutrients-15-01640-f006:**
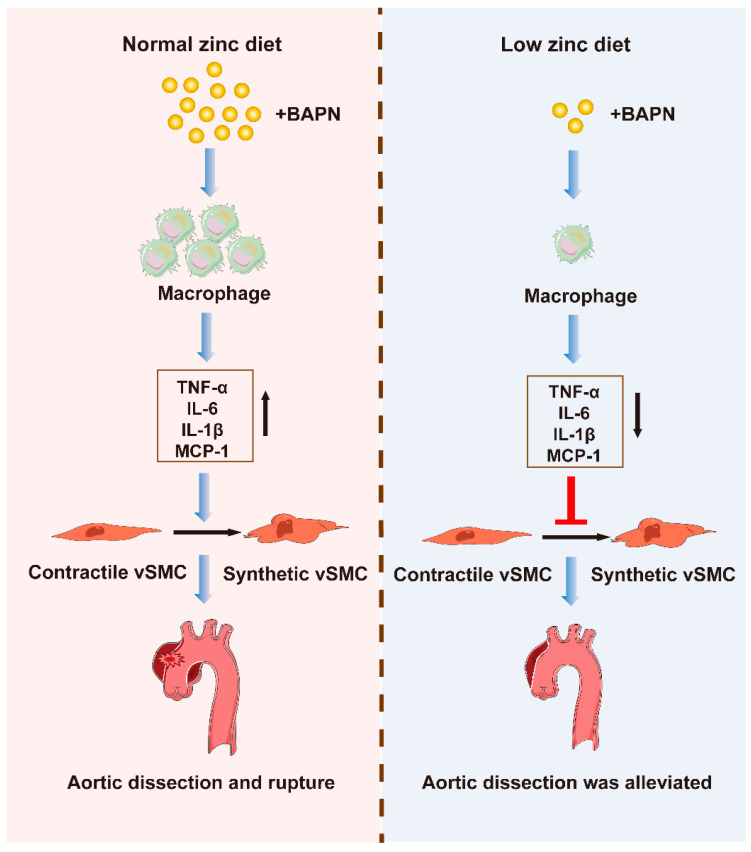
Schematic illustration of low zinc alleviated the progression of TAD. In BAPN-treated mice, compared with a normal zinc diet, low zinc treatment reduces the infiltration of macrophage, decreases the expression of cytokines and chemokines, and inhibits the phenotype switch from contractile VSMCs to synthetic VSMCs, eventually alleviating TAD development. Part of the elements in this figure uses resources from Servier Medical Art under a Creative Commons Attribution license. TAD indicates thoracic aortic dissection, and BAPN indicates β-aminopropionitrile monofumarate.

## Data Availability

The data presented in this study are available in the article or Appendix A.

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
