# Peer review of "Low Zinc Alleviates the Progression of Thoracic Aortic Dissection by Inhibiting Inflammation"

_nutrients, 2023, doi:10.3390/nu15071640_

Round 1

Reviewer 1 Report

Review comments

The research article by Lin Zhu titled “Low zinc alleviates the progression of thoracic aortic dissection by inhibiting inflammation”. Some of my concerns are:

1.     The authors mentioned that Survival curve indicated that BAPN group mice died from aortic dissection and rupture began after 20 days of BAPN administration, while BAPN + low zinc group mice began to die after 27 days. It just means that the mice died a slower death even in the low zinc treatment group.

2.     Please provide good images as I cannot evaluate the staining data properly.

3.     When did the experiment ended and the datas shown in the figures were collected at which time points of the study.

4.     The authors have just measured the expression of proinflammatory cytokines. I request the authors to evaluate more markers for prolonged sustained inflammation in BAPN groups and the reduction of inflammation in BAPN + low zinc group.

5.     The grammatical mistakes and English language has to be evaluated from native English writer.

6.     The data seems insufficient and I request to evaluate the expression of proteins as well.

Reviewer 2 Report

Congratulations on the manuscript, I will give some suggestions to improve the manuscript.

The abstract appears to provide a clear and concise summary of the study's findings and the main points of the research. However, a few additional details could be useful for a reader to understand the study better:

·       The methods used to induce TAD with β-aminopropionitrile monofumarate (BAPN) could be briefly described.

·       The dose or concentration of low zinc used in the study could be mentioned.

·       The number of mice used in each group could be provided to assess the statistical significance of the results.

·       The potential limitations or future directions of the study could be discussed. For instance, it could be interesting to investigate the optimal dose and duration of low zinc supplementation for TAD prevention and its potential adverse effects.

The introduction provides a clear overview of the problem, the importance of the study, and the rationale for investigating the role of zinc in the progression of thoracic aortic dissection. It also highlights the current knowledge gap and the potential benefits of nutritional intervention in preventing or delaying AD progression.

The statistical analysis seems appropriate and well-adjusted to the study's purpose.

The results appear to be described with great depth and clarity, suggesting that low zinc treatment can improve survival and reduce the incidence of aortic dissection in the mouse model induced by BAPN.

Some limitations should be considered:

Firstly, the study only used an in vivo mouse model, and it is unclear whether low zinc treatment would have the same effect in vitro or humans, as you mentioned. Secondly, the study only investigated the effects of low zinc treatment and did not explore the potential impact of other dietary factors or interventions.

Additionally, the study found reduced zinc levels in the arteries and serum/plasma of patients with AD based on available clinical data. However, whether zinc deficiency contributes to developing aortic dissection in humans is still unclear.

Finally, the authors acknowledge that the regulation of low zinc in AD is complex, involving immune cells, inflammation, and related factors, which means that further research is needed to understand the underlying mechanisms fully.

As a reflection, for your consideration:

Future research could focus on investigating the molecular mechanisms underlying the protective effects of low zinc on TAD development, particularly in cell experiments in vitro. Additionally, more studies are needed to explore whether the constructed conditional zinc transporter/zinc storage protein gene knockout mice have a more positive inhibitory effect on TAD development than low zinc treatment alone.

 In terms of practice, this study highlights the importance of maintaining zinc homeostasis in the body to prevent TAD development. Clinicians may consider monitoring zinc levels in patients with AD and supplementing with zinc if necessary. However, further research is needed to determine whether zinc supplementation in zinc deficiency patients can prevent or reduce the incidence of AD.
